# Polarization Switching Kinetics in Thin Ferroelectric HZO Films

**DOI:** 10.3390/nano12234126

**Published:** 2022-11-22

**Authors:** Ekaterina Kondratyuk, Anastasia Chouprik

**Affiliations:** Moscow Institute of Physics and Technology, 9 Institutskiy Lane, 141700 Dolgoprudny, Russia

**Keywords:** ferroelectric memory, ferroelectric hafnium oxide, polarization switching kinetics, switching speed, charge injection, depolarization field

## Abstract

Ferroelectric polycrystalline HfO_2_ thin films are the most promising material for the implementation of novel non-volatile ferroelectric memories because of their attractive properties, such as compatibility with modern Si technology, perfect scalability, low power consumption and high endurance. However, for the commercialization of ferroelectric memory, some crucial aspects of its operation should be addressed, including the polarization switching mechanism that determines the switching speed. Although several reports on polarization switching kinetics in HfO_2_-based layers already exist, the physical origin of retardation behavior of polarization switching at the low and medium switching fields remains unclear. In this work, we examine several models of switching kinetics that can potentially explain or describe retardation behavior observed in experimental switching kinetics for Hf_0.5_Zr_0.5_O_2_ (HZO)-based capacitors and propose a new model. The proposed model is based on a statistical model of switching kinetics, which has been significantly extended to take into account the specific properties of HZO. The model includes contributions of the depolarization field and the built-in internal field originating from the charge injection into the functional HZO layer during the read procedure as well as in-plane inhomogeneity of the total electric field in ferroelectric. The general model of switching kinetics shows excellent agreement with the experimental results.

## 1. Introduction

Since the discovery of ferroelectric properties in doped (or alloyed) hafnium oxide polycrystalline thin films [1], this material is attracting the interest as a promising functional material for nonvolatile ferroelectric memories due to its perfect compatibility with modern Si technology. A number of excellent performances have been demonstrated, including full scalability, low power consumption, high endurance, and nanosecond switching speed, though some challenges have still not been overcome and they hinder the commercialization of HfO_2_-based memory.

These challenges originate from the inherent property of ferroelectric, specifically, from the omnipresent depolarization field produced in ferroelectric film by the surface charge of spontaneous polarization. At the poled state, the depolarization field of the polarization charge induces the accumulation of non-ferroelectric charges of the opposite sign at capacitor interfaces, which, in turn, produce the built-in electric field aligned against the depolarization field [2,3]. During the readout procedure, the built-in field partially offsets the applied field and, thus, real field in ferroelectrics turns out to be smaller than the expected field. For a part of domains, the real field occurs smaller than the coercive voltage; these domains do not switch during the readout procedure after long-term information storage, which means polarization loss and could cause readout failure. Another undesirable consequence of the offset of the applied field is a decrease in the switching speed [4]. Indeed, as it is predicted by laws of polarization switching kinetics, the polarization switching speed depends on the electric field in the ferroelectric [5,6,7,8,9], and this is a fundamental property of any ferroelectric film.

Several models giving a relation between the electric field in the ferroelectric film and the time needed for the switching event to occur have been proposed. For a long time, the Kolmogorov–Avrami–Ishibashi model (KAI) [5,6,7] was the main model of the polarization switching kinetics. It describes polarization reversal as the formation of a nucleus, which expands through the movement of the domain walls until the whole film is switched in the opposite polarization state. The KAI model is in a good agreement with experimental switching kinetics for single crystals and epitaxial films. However, it cannot be applied to polycrystalline ferroelectric thin films, because in such films switching takes place over a wide interval of switching times, which contradicts the predictions of the KAI model.

To explain polarization switching kinetics of polycrystalline films, several approaches has been proposed. The inhomogeneous field mechanism (IFM) model encompasses the inhomogeneous distribution of the electric field in polycrystalline film across the electrode area and helps to describe the various structural non-uniformity of the polycrystalline film [10]. For ferroelectric HfO_2_, by means of the IFM model, it has been demonstrated that polarization switching accelerates and field distribution shrinks during the training process of the as-prepared film (the so-called wake-up effect) [11]. An alternative way to consider the impact of the polycrystalline microstructure is based on phase-field modelling, which showed that the grain pattern (e.g., grain size and orientation) and the existence of grain boundaries affect the polarization switching or domain evolution dynamics in ferroelectric nanofilms [12,13].

The most widely used model to describe the experimental polarization switching kinetics of polycrystalline films is a nucleation-limited switching (NLS) model [8]. In the NLS model, the ferroelectric film consists of numerous areas (domains), each with its own independent switching time. The time needed for the switching of an elementary region (switching time) is equal to the waiting time for the nucleation of a reversed domain, i.e., the time needed for filling the region with the expanding domain is neglected compared to the waiting time. Switching times are distributed in an exponentially wide time interval. It is suggested that the individually switched regions correspond to single grains or clusters of grains, in which the grain boundaries act as frontiers limiting the propagation of the switched region. The NLS model was used to predict the polarization loss in Al-doped HfO_2_ films caused by thermal depolarization upon information storage [9]. The NLS model has also been employed to study the impact of different parameters of HfO_2_ on its switching speed. In particular, it has been shown that concentration of oxygen vacancies [14,15], HZO film thickness [16], and Zr alloy concentration in Hf_1−x_Zr_x_O_2_ thin films [17] affect polarization switching kinetics. On the other hand, the impact of several physical phenomena in HfO_2_ on time dependence of polarization switching has been examined in the frame of the NLS model. In particular, it has been shown that it depends on local field inhomogeneity [17] and charge injection into the functional HZO layer [4]. For other (non-hafnia) polycrystalline ferroelectric films, by means of the statistical model of switching kinetics, it has been reported that the depolarization field also contributes to polarization switching [18]. This approach has not been applied to HfO_2_ films, possibly due to the fact that in the case of HZO no depolarization is observed at zero applied voltage [19]. 

Therefore, several models have been used to describe the switching kinetics of polycrystalline ferroelectric HfO_2_ films [20], but the domain reversal behavior at medium and low applied fields has not been addressed yet; specifically, the observed retardation of polarization switching by long switching pulses of small and medium amplitudes is still non-clear. Meanwhile, due to the built-in electric field emerged upon information storage, during the readout procedure the applied field is offset and polarization is switched by the low real field. Therefore, for successful engineering of ferroelectric capacitors for memory applications, it is necessary to understand the physical mechanism behind the retardation of polarization switching at low field and to find an effective model to accurately describe experimental switching kinetics. In this work, we examine models of polarization switching kinetics that are able to describe retardation behavior of polarization switching by medium and low fields and propose a general model that takes into account the specific properties of ferroelectric HfO_2_ films; specifically, a model includes contributions of the depolarization field and the built-in internal field originating from the charge injection as well as in-plane inhomogeneity of the total electric field in ferroelectric. A general model of switching kinetics shows excellent agreement with the experimental results for HZO 10 nm in thickness, which is now recognized as the most promising material for back-end-of-line (BEOL) ferroelectric random access memory (FeRAM) [21].

## 2. Materials and Methods

The studies were carried out using ferroelectric Hf_0.5_Zr_0.5_O_2_ (HZO)-based capacitors 25 μm in size. The HZO (10 nm) layer was grown via the thermal atomic layer deposition technique at a 240 °C substrate temperature. W (40 nm) and TiN (20 nm)/Al (150 nm) films grown by magnetron and electron beam sputtering served as the bottom and top electrodes, respectively. Crystallization of the functional HZO layer into the ferroelectric structural phase was carried out by annealing for 30 s at 500 °C in an Ar. The wake-up procedure was carried out by 10^5^ switching bipolar cycles of pulses of 3 V/100 µs, refreshing was carried out by 10 cycles of the same pulses. The remanent polarization was 25 μC/cm^2^, the average coercive voltages were −1.1 and 1 V. RC time constant was less than 50 ns.

The datasets on switching kinetics were measured by applying to the top electrode the sequence of pulses shown in Figure 1a with simultaneous reading of the transient current from the bottom electrode. Before the pulse sequence, a number of bipolar cycles were applied in order to wake-up the as-prepared HZO film or to refresh the state of the structure after measurement of the previous experimental point. A preset pulse of 3 V/10 µs switched the structure into a single-domain state. A set pulse of opposite polarity had amplitude in the range from 0.9 V to 3 V and a duration from 30 ns to 0.3 ms. The duration of pulse trails was chosen to be 100 ns in order to minimize their contribution to polarization switching. Set pulse partially switched the polarization, which was then measured by applying two pulses of 3 V/10 µs (read pulses). Subtracting the current response of the structure to the fourth pulse from the current response to the third pulse allowed to calculate the response corresponding directly to the polarization reversal back to the single-domain state. In this way, we determined the fraction of the polarization switched during the set pulse. The amplitude and duration of the read pulses were chosen to ensure the complete switching of the polarization (3 V/10 µs). To avoid emergence of the built-in field, delays between all pulses were kept equal to 50 ns [4].

Since accurate examination of the models requires highly reliable experimental data on switching kinetics, we considered the contribution of set pulse trails to switched polarization. We roughly estimated what fraction of the polarization would be switched by a set pulse (which is, in fact, a trapezoid) if it were rectangular, while its length would be equal to the length of the trapezoid plateau. For this purpose, we calculated the root mean square voltage *V*_RMS_ corresponding to the applied set pulse. We corrected the experimentally read polarization value by multiplying it by the ratio *V*_RMS_/*V_set_*. As it should be expected, a smaller fraction of polarization was switched than in the experiment, and the contribution of 100-ns-long trails to switched polarization was significant for short set pulses and minor for long set pulses (Figure 1b).

## 3. Results and Discussion

### 3.1. Examination of NLS Model

In the first place, we applied the classical NLS model to the corrected experimental data. According to this model, the fraction of the switched ferroelectric film *P*(*t*) is related to the distribution function of switching times *g* (*z*) by following expression [8]: (1)$$P\left(t\right)=\int_{-\infty }^{\ln t}g\left(z\right)\;dz$$
where $P\left(t\right)=1-\frac{\sum_{NS}S_{i}}{\sum_{all}S_{i}}$ (NS means non-switched domains) and the distribution function meets the normalizing condition $\int_{-\infty }^{\infty }g\left(z\right)dz=1$, z = log *t*_0_. Here, *S*_i_ is the area of *i*-th domain and *t*_0_ is unique waiting time of the elementary regions.

In the classical application of the NLS model, its authors propose the use of a distribution function of the switching times, which is flat for z lying between z_1_ and z_2_ ($t_{\mathrm{left}}=10^{z_{1}}$ and $t_{\mathrm{right}}=10^{z_{2}}$ stand for the left and right limits of the flat part of the switching times spectrum):(2)$$g\left(z\right)=\frac{\Gamma^{2}h}{{\left(z-z_{1}\right)}^{2}+\Gamma^{2}}\;\;\;\mathrm{at}\;z\;<z_{1},\;g\left(z\right)=h\;\;\;\mathrm{at}\;\;z_{1}<z<z_{2},\;g\left(z\right)=\frac{\Gamma^{2}h}{{\left(z-z_{2}\right)}^{2}+\Gamma^{2}}\;\;\;\mathrm{at}\;\;z_{2}<z,\;h={\left(z_{2}-z_{1}+\Gamma \pi \right)}^{-1}.$$

Here, *h* and Г are fitting parameters for the amplitude and the rate of decay of the distribution function, respectively.

Fitting function (2) produces linear dependence of the switched polarization on log *t*_set_ in a medium range of polarization values. Full dependence of the switched polarization on log *t*_set_ is similar to the arctan function (solid lines in Figure 1c). Meanwhile, experimental dependences are significantly non-linear in a medium range of switched polarization (dotted lines join experimental points and serve as a guide for eyes). Thus, classical implementation of the NLS model does not accurately describe experimental switching kinetics in HZO.

One of the possible reasons for the demonstrated inaccuracy of the NLS model for HZO thin films is that the model does not take into account the restrictions on the maximal possible fraction of switched domains at a particular applied voltage. The corresponding maximal polarizations for each voltage can be found from the area of the limited *I–V* curve between zero voltage and the selected one, or from the *P–V* curve, as shown in Figure 2a. As can be seen from Figure 2b, this limitation makes it possible to significantly improve the quality of the model curves. The resulting distributions of switching times are shown in the lower part of Figure 2b. 

An alternative distribution function of switching times *g*(log *t*_0_) in the NLS model is the Lorentzian distribution function *F*(log *t*_0_) (3) of the switching times of the domains *t*_0_ (time for nucleation) [22]:(3)$$F\left(\log t_{0}\right)=\frac{A}{\pi }\left[\frac{\omega }{{\left(\log t_{0}-\log t_{Lorentz}\right)}^{2}+\omega^{2}}\right].$$

Substituted into following expression, the Lorentzian distribution function gives the dependence of the switched polarization on *t*: (4)$$\Delta P\left(t\right)=2P_{s}\int_{-\infty }^{\infty }\left[1-\exp\left\{1-{\left(\frac{t}{t_{0}}\right)}^{n}\right\}\right]\;F\left(\log t_{0}\right)d\left(\log t_{0}\right).$$

Here, *t_Lorentz_* is the position of the maximum of the Lorentzian function, *A* is a norma-lization constant, *ω*(log *t_Lorentz_*) is the half-width at half-maximum, and *n* is the factor related to the dimension of ferroelectric material (*n* = 1 for thin film).

From Figure 2c, it is evident that the NLS model with Lorentzian distribution of switching times describes full switching kinetics, including the retardation behavior of polarization switching by low and medium fields (for this model, we also apply the restriction on the maximal switching of polarization by the selected voltage). To explain the observed Lorentzian distribution of log *t*_0_, Jo et al. assumed that a local field exists at the ferroelectric domain pinning sites and that it has a Lorentzian distribution [22]. The authors also concluded that at low external field *E* position of Lorentzian distribution satisfies the following relation:(5)$$\log t_{Lorentz}\approx \frac{\alpha }{E},$$
where α is the activation field, i.e., the electric field that would cause immediate polarization switching. In previous work [4], we indeed obtained a linear dependence of *t_Lorentz_* on voltage in the range of small voltages, which is consistent with the results obtained in [22]. However, as can be seen from the plot in Figure 2d, the dependence obtained in this work for wider range of voltage is more similar to that obtained using the classical distribution function (2) for the NLS model [8]. It can be seen that the *t*_Lorentz_ are located exactly between the *t*_left_ and *t*_right_. Thus, both kinds of NLS models give the same result. 

It should be noted that the flat distribution of switching times from the classical NLS model [8] has very large tails related to large Г = 0.8. Each of the tails is equal to three orders of magnitude of switching times, i.e., two tails take six orders, while the flat part of the distribution function is equal to just one order (Figure 2b,c). Thus, the classical flat distribution turns out to be very similar to the Lorentzian distribution, and, therefore, the classical NLS model transforms by itself to the NLS model with the Lorentzian distribution.

Characteristic switching times *t_Lorentz_* (as well as *t*_left_ and *t*_right_) were extrapolated to high voltages (Figure 2d) by function [8]:(6)$$t_{Lorentz}=t_{\inf}\exp\left(\frac{V_{set}}{V_{\alpha }}\right),$$
where *t*_inf_ is characteristic switching time (i.e., the shortest time necessary for switching) and *V*_α_ is voltage corresponding to the activation field α. The parameters of fitting are summarized in the table at the end of this work. Extrapolation makes it possible to predict the kinetic behavior at different voltages, in particular, to reconstruct the distribution functions and switching polarization curves. For example, fitting of switching times gives the mean switching time (position of Lorentzian distribution) of 20 ns for *V_set_* = −2.5 V (inset in Figure 2d), whereas it is difficult to get appropriate experimental points directly, because this voltage causes almost complete polarization switching in the range of available *t*_set_. At *V_set_* = −5 V (which is a bit larger than the breakdown voltage 4.7–4.8 V) it decreases to 2 ns, while the characteristic switching time is 0.1 ns. Similar switching times, including characteristic switching time, were obtained for any *V_set_*.

It should be noted that Jo et al., who proposed the Lorentzian distribution of switching times, observed non-exponential behavior of *t*_Lorentz_ on *V_set_*. We assume that the difference with this work may be related to the contribution of the set pulse trails to the polarization switching taken into account in current work. Alternatively, this could indicate that the NLS model with the Lorentzian distribution function of switching times may not physically explain the retardation of polarization switching at low fields, although it can serve as a convenient tool for the fitting of experimental switching kinetics. 

### 3.2. Impact of Charge Injection during Set Pulse

An alternative explanation of retardation behavior may come from charge injection and charge accumulation in the nearby-electrode passive layer of the ferroelectric film. During the long set pulse, due to field-induced charge injection, the built-in field emerges and increases with time, which causes the offset of applied voltage *V*_appl_ and decrease in the real voltage *V*_f_ right during the time of switching [4]. To calculate the instant real voltage *V*_f_, we numerically solved the following system of Equations (2)–(4):(7)$$d_{d}E_{d}+d_{f}E_{f}=V_{\mathrm{appl}},$$
(8)$$\varepsilon_{d}E_{d}+\varepsilon_{f}E_{f}=\frac{\sigma +P_{\max}}{\varepsilon_{0}},$$
(9)σ=jt.

Here, *d*_d_ and *d*_f_ are the thickness of dielectric and ferroelectric layers, respectively (*d*_d_ + *d*_f_ = *d* = 10 nm is total thickness of HZO), *E*_d_ and *E*_f_ are electric fields across these layers, ε_d_ and ε_f_ are their dielectric constants, ε_0_ is the vacuum permittivity, *V*_appl_ is external applied voltage, *P*_max_ is switchable polarization, σ is surface density of the injected charge, *t* is time of injection, and *j* is the current density across interface dielectric layer. To calculate current transport, we used the thermionic emission law with Schottky contribution:(10)$$j=A^{*}T^{2}\exp\left[-\frac{e\left(\varphi_{B}-\sqrt{eE_{d}/4\pi \varepsilon_{r}\varepsilon_{0}}\right)}{kT}\right],$$
where *A** is the effective Richardson constant, *T* is the absolute temperature, *e* is the elementary charge, $e\varphi_{B}$ is the Schottky barrier height (i.e., conduction band offset), *k* is the Boltzmann constant, and ε*_r_* is the optical dielectric constant.

The temporal dynamics of injected charge σ and real temporal profile of electric field across ferroelectric during passing the set pulse are shown in Figure 3b (the algorithm for determining the parameters needed in (6)–(8) is given in previous work [4]). It is evident that during the application of long set pulses the decrease in real electric field *E*_f_ is more prominent. Moreover, the relative decrease in real field (normalized to maximal field before the start of the charge injection) is more prominent for the low applied electric field.

To examine the impact of charge injection on the retardation behavior, we calculated the real electric field for set pulses of different duration at 25 and 125 °C (Figure 3c). At 125 °C, real voltage drops from 1.6 V at the start of voltage application to 0.95 V at the end of a 0.5-ms-long pulse. This means that each of the right-hand experimental points belongs to different classical NLS curves corresponding to a different set voltage (hypothetical curves are schematically shown in Figure 3c). This effect may manifest itself in an apparent retardation of polarization switching at the low electric field and explain deviation of kinetics data from the classical NLS model. It is noteworthy that in this case, the Lorentzian distribution of switching times suits well, but it does not reflect the physics behind retardation behavior, and just play a descriptive role.

The change in real field at 25 °C is found to be smaller than at 125 °C (1.16 V after 0.5 ms from the start of 1.6 V application). Meanwhile, the experimental dataset at both room and elevated temperatures are qualitatively similar, and no changes in the retardation rate are observed. Thus, we can conclude that charge injection alone cannot explain the retardation behavior and other roots should be analyzed.

### 3.3. General Model of Polarization Switching Kinetics

The so-called statistical model of switching kinetics [18] was taken as the basis of the general model of switching kinetics. Initially, this model was developed by Lou to explain the observed retardation behavior of polarization switching in polycrystalline ferroelectric film at the low applied field, i.e., in terms of the NLS model, the statistical model could explain large (three-orders-of-magnitude) tails of switching time distribution. This model is based on taking into account the time-dependent depolarization field *E*_dep_. In this model, the depolarization field is averaged over all ferroelectric film and, thus, it depends on the fraction of switched domains, which, in turn, is determined by pulse duration (at some applied field). The model is based on feedback equations:(11)$$\frac{M_{0}-N-1}{M_{0}-N}=\exp\left(-\frac{t_{N+1}}{t_{\inf}}\exp\left(-\frac{\alpha }{E_{\mathrm{appl}}+E_{\mathrm{dep}}}\right)\right)$$
(12)$$E_{\mathrm{dep}}=\frac{d_{d}P_{M_{0}-N}}{d_{f}\varepsilon_{d}\varepsilon_{0}}$$
(13)$$P_{M_{0}-N}=\frac{M_{0}-2N}{M_{0}}P_{M_{0}}$$
where *M*_0_ is the total number of domains, *N* is the number of switched domains at the time moment *t*_*N*+1_, *t*_inf_ is the characteristic switching time (switching time at infinite applied field), α is the activation field, and *E*_appl_ is the applied field. From Formula (12), it is evident that the depolarization field is non-zero only if the structure contains a dead layer at the electrode interface, as it is shown above in Figure 3a.

In the first place, we examined whether domain switching takes place in a random way, as it is implied in the statistical model, or in a predetermined way. Using the piezoresponse force microscopy (PFM) implemented in the AFM Ntegra (NT-MDT Spectrum Instruments) [23], we mapped the domain structure of the HZO capacitor after five identical switchings. The procedure of switching is the same as for the acquisition of the experimental point in the kinetics dataset, except for the application of the read pulses. Instead of them, the PFM mapping was made. As it can be seen from Figure 4a, the domain switching is generally predetermined, i.e., switching predominantly takes place in certain regions, although some sporadic switching is also observed.

The domain structure had no correlation with the topography of studied area (Figure 4c), which is reasonable. Indeed, grains that are visible at the surface of the ferroelectric capacitor are the grains of the top TiN electrode, whereas the domain structure is determined by the whole set of properties of the capacitor, and domain switching takes place inside the capacitor. Therefore, topography could not help to resolve the role of HZO grain patterns in polarization switching. However, the PFM amplitude map gives some information. Amplitude is proportional to the local piezoelectric coefficient and it reflects the magnitude of the out-of-plane component of the polarization vector. That is, in Figure 4b,d, the white color corresponds approximately to the out-of-plane (*z*) orientation of the polarization vector, whereas the black color approximately stands for the in-plane (*xy*) orientation. After full switching of polarization, the distribution of the out-of-plane component of polarization over the studied area could be seen (Figure 4d). Comparison of this map with the domain structure after switching of the small part (~15%) of polarization (Figure 4a,b) indicates that switching first takes place in the region with the most vertical component of polarization. This is reasonable because one could expect that such regions have the smallest coercive voltage. It should be noted that the predetermined character of polarization switching revealed in 10 nm-thick HZO should be additionally examined for films with other parameters, for example, for films with other thickness, other average size of grains, and other structural texture.

The predetermined character of polarization switching could be explained by spatial distribution of local electric fields originating from defects distribution, grain pattern, and local enhancement/weakening of the applied electric field due to hills/pits at the surface of the bottom electrode. To simulate and take into account this issue in the model of switching kinetics, for each domain out of *M*_0_, we took *d*_f_ as *d* − *d*_d_, where *d*_d_ varies from 0.4 nm to 0.8 nm with a Lorentzian distribution (in Figure 4b, sixteen domains with different *d*_f_ are shown).

The second phenomenon that could contribute to the polarization switching is the charge injection during the set pulse. In order to take it into account in the general model, we calculated *E*_f_ using Equations (7)–(10) and substituted it into Equation (11) instead of *E*_appl_. Since *E*_f_ = *V*_f_/*d*_f_, we obtained the Lorentzian distribution of the *E*_f_ in *M*_0_ domains (Figure 5a,d). The temporal evolution of *E*_f_ calculated for each domain (the total number of which was taken as *M*_0_ = 1000) in the time interval from 0.1 ns to 10 ms is shown in Figure 5a.

Third, the formula for the depolarization field should be modified to meet the specific properties of ferroelectric hafnium oxide. Previously, we have shown that the depolarization does not contribute to the retention loss in the HZO capacitor, i.e., no depolarization is observed at zero applied voltage upon the information storage [19]. Possibly, this fact is associated with the effective screening of polarization charges by non-ferroelectric charges, of which volume density reaches 10^21^–10^22^ cm^−3^ in HZO [24]. On the other hand, at non-zero applied voltage, the effectiveness of screening should deteriorate, and the non-screened depolarization field could cause the retardation behavior of switching kinetics as it was predicted by Lou. Following the recent work on the depolarization field [25], we normalized the depolarization field *E*_dep_ to the relation of the applied voltage to coercive voltage *V*_c_:(14)$$E_{\mathrm{dep}}=\frac{d_{d}P_{M_{0}-N}}{d_{f}\varepsilon_{d}\varepsilon_{0}}\cdot \frac{V_{\mathrm{appl}}}{V_{c}},$$

Therefore, the total electric field in the ferroelectric layer consists of three components: the applied electric field, the field produced by the charge injected upon the set pulse, and the depolarization field, and they are all varied by the Lorentzian distribution to simulate the spatial distribution of local electric fields.

In addition, it is important to note that the proposed model takes into account which maximal polarization can be switched by the applied voltage of a particular amplitude, as it is described above. 

Results of the simulation of switching kinetics within the statistical model are in good accordance with the experimental data (upper plots at Figure 5b,c for 25 °C and 125 °C), and this seems to be reasonable because the calculated depolarization field reached ~1.2 MV/cm in the single-domain state of HZO (Figure 4e), i.e., it was close to coercive voltage. As can be seen from Formula (12), the depolarization field does not depend on temperature, so plots in Figure 5e are almost the same for 25 °C and 125 °C and just shrink along the time axis. In Figure 5b,c, the central plots represent the total field in the ferroelectric, which is obtained by adding the fields from the lower plots (obtained from Equations (7)–(10)) and the depolarization field (Figure 5e). The injected charges screen the polarization charge and at elevated temperature, the rate of the charge injection is larger, so in a result, one could expect the suppression of the retardation behavior of polarization switching at elevated temperatures (Figure 5c).

Looking for other possible reasons for depolarization-like impact on switching kinetics, we calculated the distribution of the vector electric field in HZO at the polydomain state. Indeed, the specific feature of the statistical model of Lou is that the shape of kinetics curves strongly depends on the fraction of switched domains, i.e., the retardation behavior is determined by this fraction. The electric field produced by domain walls in the polydomain state was calculated by means of finite element analysis (Comsol Multiphysics 5.4 software). The geometry of the model is shown in Figure 6a; we took HfO_2_ (10 nm) from the Comsol database of materials for HZO and set size of domains equal to 50 nm. To simulate the field of spontaneous polarization, we used Electrostatics physics of Comsol and placed positive and negative uniform surface charge σ_P_ at the top and bottom surfaces of HZO. The magnitude of surface charge σ_P_ was chosen to be equal to 2 μC/cm^2^, because we have previously found that such polarization surface charge corresponds to similar remanent polarization [26]. Simulation reveals that at domain walls, the in-plane component of the electric field of polarization charges is similar and even larger than the out-of-plane component, which is usually considered as the depolarization field (Figure 5a). During applying an external field, the electric field of domain walls could locally offset it and, thus, cause the retardation of switching. At a first glance, it may seem that in this case, the retardation would depend not on the fraction of switched domains but on the length of the domain wall. Indeed, as it is schematically shown in Figure 5b, during switching to the upward polarization state, the contribution of the electric field of the domain wall to the switching of the neighbor region of concave domain wall is large. However, the number of concave regions of domain walls should be equal to the number of convex regions, in which the contribution is minor. Thus, the impact of bended (and even highly fractal) domain walls is equivalent to the impact of straight domain walls, i.e., to the fraction of the switched domain.

In the Table 1, polarization switching parameters of ferroelectric capacitors based on hafnium oxide are summarized. The table presents the results of this, and other works obtained by various methods, such as the NLS model, IFM model, and the proposed general model. As can be seen, for all samples based on Al doped HfO_2_, *t*_inf_ = 1 × 10^−13^ s, while for samples based on HZO, *t*_inf_ takes values of the order of 10^−10^ s. However, it should be noted that for capacitors of the first type, only the results obtained by the NLS method are considered. As for the activation energy, its values are of the same order and range from 9 MV/cm to 20 MV/cm for all samples and methods.

## 4. Conclusions

In summary, we examined models of polarization switching kinetics that can explain or describe retardation behavior observed in experimental kinetics in HZO capacitors. We conclude that the NLS model with the Lorentzian distribution of switching times is the most convenient tool to fit experimental results and, thus, it can be used for tasks that require the prediction of switching kinetics under different conditions, for example, for the calculation of the retention loss upon long-term information storage. However, the physics behind the Lorentzian distribution of switching times is non-evident because its right (long-time) fragment could be produced by different phenomena. For example, it can be due to the charge injection and charge accumulation in the nearby-electrode passive layer of the HZO film. 

To take into account several physical phenomena affecting the polarization reversal in hafnium oxide, we proposed a general model based on equations of Lou’s statistical model of switching kinetics. We made three major changes. First, in order to simulate the defects contribution and effect of the topography of the bottom electrode, we introduced the inhomogeneity of total field across the electrode area in the model. Second, the depolarization component was corrected, taking into account the screening effect in hafnium oxide film. The depolarization field became zero at zero applied voltage and increased with applied voltage. Finally, we added the contribution of the electric field produced by charge injected into the interfacial layer of HZO during the readout procedure. The proposed general model is in good agreement with the experimental data for HZO capacitors. Unlike the NLS model, the proposed model allowed to explain physics behind retardation behavior of the polarization switching by medium and low applied fields.

## Figures and Tables

**Figure 1 nanomaterials-12-04126-f001:**
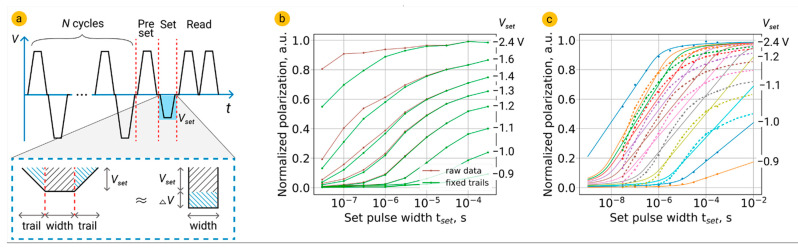
(**a**) The sequence of pulses used to obtain data on switching of the polarization depending on the amplitude and duration of the set pulse. Below: impact of set pulse trails to switched polarization. (**b**) Comparison of raw and corrected data. (**c**) Result of fitting of experimental dataset (points) by the classical NLS model (solid lines) and the expected shape of the curves (dotted lines).

**Figure 2 nanomaterials-12-04126-f002:**
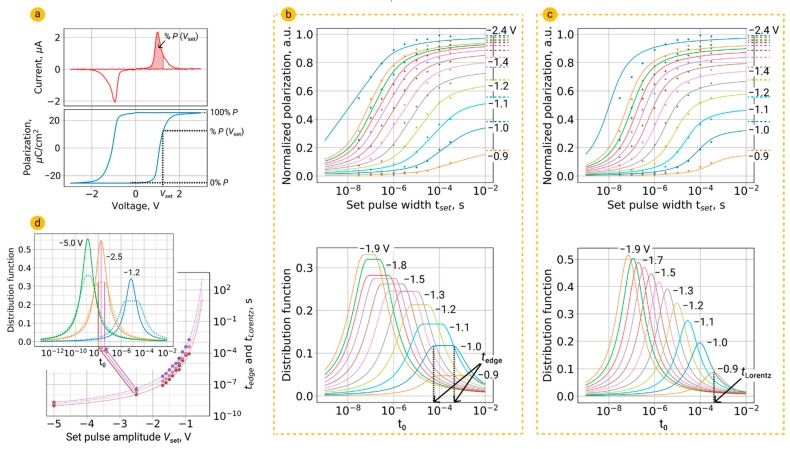
Examination of NLS model with flat and Lorentzian distributions of switching times. (**a**) Switching *I–V* and *P–V* curves. (**b**) Result of fitting by the NLS model with a set limit on the maximal switching by each voltage; distributions of log *t*_0_. Dotted lines notate maximal possible switching polarization for different *V_set_*. (**c**) Result of fitting by the NLS model with the Lorentzian distribution of switching times (with a set limit on the maximal switching by each voltage); Lorentzian distributions of log *t*_0_. (**d**) Comparison of the voltage dependences of the *t*_left_ (brown) and *t*_right_ (lilac) and *t*_Lorentz_ (pink). The inset shows the positions of the predicted peaks of both kinds of distributions for unknown voltages.

**Figure 3 nanomaterials-12-04126-f003:**
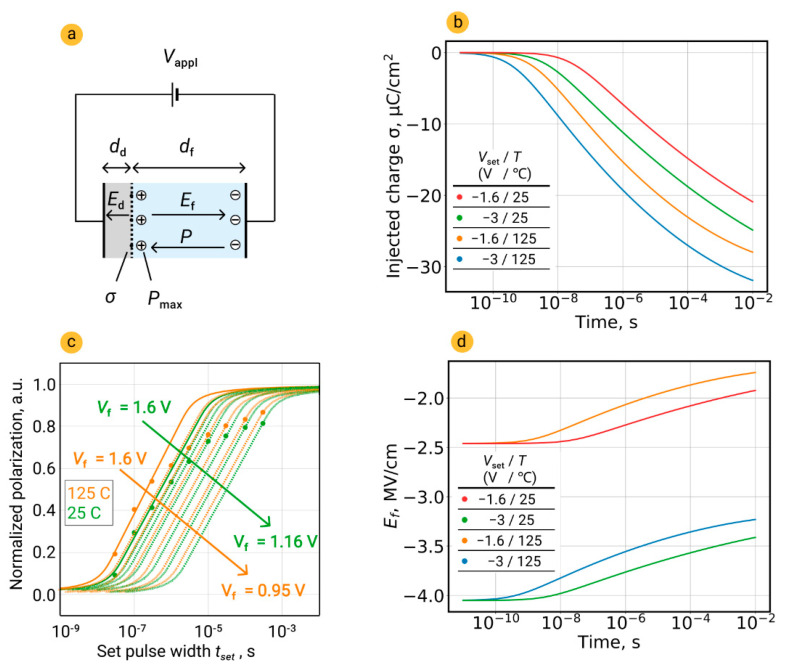
Impact of charge injection on retardation behavior of polarization switching kinetics. (**a**) Sketch of the electroded structure consisting of the interfacial dielectric and ferroelectric layers. (**b**,**d**) Temporal dynamics of injected charge and real profile of electric field across ferroelectric during passing the set pulse. (**c**) Examination of the impact of the charge injection during the set pulse on the retardation of polarization switching. Dots are experimental points; solid lines are simulation curves from Equations (7)–(10).

**Figure 4 nanomaterials-12-04126-f004:**
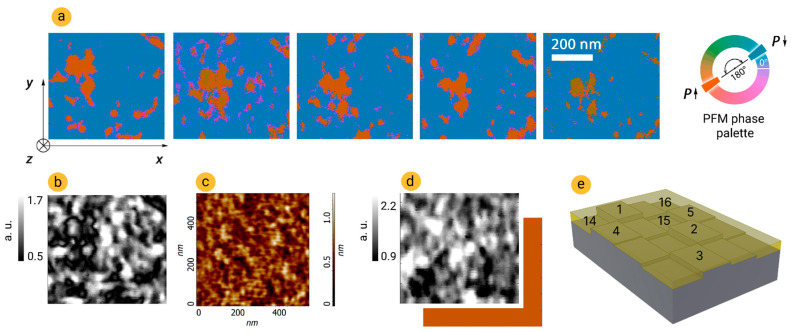
Examination of the statistical domain switching. (**a**) Domain structure (PFM phase maps) after five identical switchings by set pulse −1 V/100 μs. (**b**) PFM amplitude map and (**c**) topography of top electrode surface corresponding to PFM phase map above it. (**d**) Domain structure (overlapped PFM amplitude and phase maps) at monodomain state after applying −3 V. (**e**) Schematic representation of the domain structure of a ferroelectric with a variation in ferroelectric and dielectric thicknesses. The numbers denote the sequence of domain polarization switching.

**Figure 5 nanomaterials-12-04126-f005:**
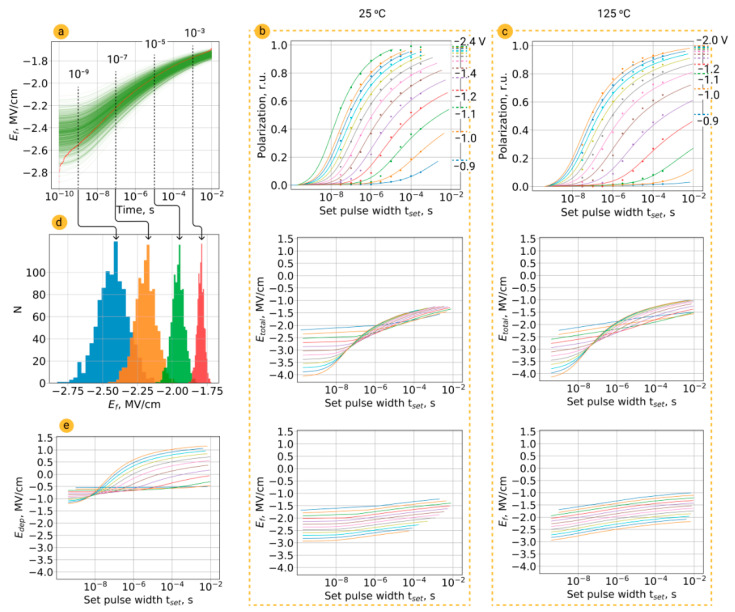
Examination of the general model of switching kinetics. (**a**) *E*_f_(*t*) dependence, taking into account the contribution of charge injection for *M*_0_ = 1000 domains with the Lorentzian distribution of the ferroelectric thickness. (**b**,**c**) Results of fitting of experimental datasets for 25 °C and 125 °C by the general model (solid lines). (**d**) Distribution of fields over *M*_0_ = 1000 domains at different time moments. (**e**) Dependence of depolarization field on time for 125 °C.

**Figure 6 nanomaterials-12-04126-f006:**
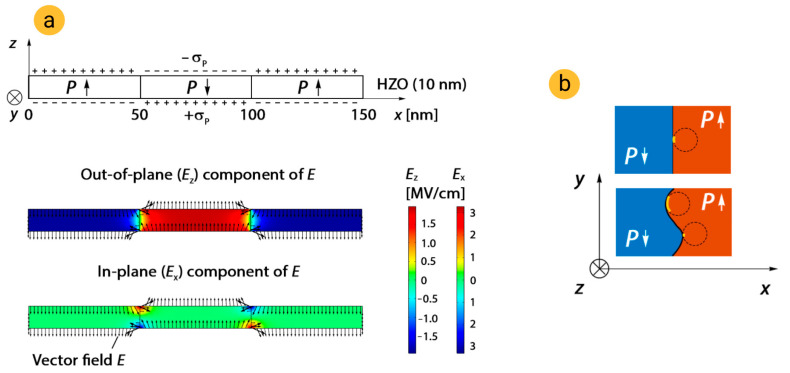
Examination of possible contribution of the electric field of domain walls. (**a**) Model parameters and results of computing of distribution of electric field. (**b**) Illustration of impact of bending and straight domain wall. Circles mark the regions, which will be switched to *P*↓ state; yellow lines mark contact segments.

**Table 1 nanomaterials-12-04126-t001:** Switching kinetics parameters for HfO_2_-based capacitors obtained by different models.

	t_inf_	α	Film	Model, Ref.
1	1 × 10^−10^ s	10–15 MV/cm	HZO (10 nm)	Classical NLS, this work
2	1 × 10^−10^ s	13 MV/cm	HZO (10 nm)	NLS with Lorentzian distribution, this work
3	4 × 10^−10^ s	18–20 MV/cm	HZO (10 nm)	General model, this work
4	1 × 10^−10^ s	8.9 MV/cm	HZO (8.9 nm)	IFM [11]
5	1 × 10^−13^ s	12.7 MV/cm	Al-doped HfO_2_ (10 nm)	Classical NLS [9]
6	1 × 10^−13^ s	11.3 MV/cm	Al-doped HfO_2_ (8 nm)	Classical NLS [27]

## Data Availability

Not applicable.

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
