# Peer review of "Polarization Switching Kinetics in Thin Ferroelectric HZO Films"

_nanomaterials, 2022, doi:10.3390/nano12234126_

Round 1
Reviewer 1 Report
The authors of this manuscript presented a comparative study of the models that are utilized to describe the polarization switching behaviors of polycrystalline HZO nanofilms. They focused on the examination of the effectiveness of these models in explaining the retardation behavior observed in experimental switching kinetics for 10 nm thick HZO-based capacitors at low field. They proposed an extended model based on Lou’s statistical model of switching kinetics in the manuscript as well. This manuscript is well presented. The authors are invited to address the following issues:
(1) I understand that the highlight of this study is the improved Lou’s statistical model which now takes into account the field inhomogeneity, depolarization field and charge injection. These are the aspects that are already believed to result in time dependence of polarization switching kinetics. What the authors did is to collect these formulations and to make the statistical model work for the observed low-amplitude switching behaviors. Of course it is always nice to see a systematic examination of the available options for modeling. It made me wonder what else novel contribution can be found through this investigation. Are there any other key mechanisms that can potentially be associated with such switching dynamics? Can this study lead to further insight into the polarization switching other than the aspects mentioned above? Is it really necessary to provide so many details associated with the models in 3.1 and 3.2? The authors have already published their studies regarding these two models, e.g., Ref. [4]. The contents in 3.1 and 3.2 are mostly identical to that published paper.
(2) The domain structures are shown in Fig. 4. However I did not see any grain morphology graphics of the HZO nanofilms. Apparently, the authors did not consider the polycrystalline microstructure that will affect the domain switching kinetics. It has been shown via phase-field modeling that the nature of the grain pattern in the nanofilms (e.g., grain size and orientation) and the existence of grain boundaries will affect the polarization switching or domain evolution dynamics in ferroelectric nanofilms, e.g., Su et al., Acta Materialia 87: 293-308, 2015; Zhang & Su, Int. J. Solids Struct. 254: 111939, 2022. The authors should discuss such findings in the context. In fact, as this study is regarding the modeling of polarization switching kinetics, the authors should widen the spectrum of their literature review by discussing microstructure-based approaches such as phase field modeling, which has been shown capable to describe time-dependence of polarization switching in ferroelectric nanofilms.
(3) The extended statistical modeling of the polarization switching is based on a predetermined switching domain as demonstrated in Fig 4(a). How are the authors so sure that this assumption is applicable to all other polarization switching kinetics of polycrystalline HZO nanofilms? How reliable is assumption? What happens if other conditions change, say the thickness, the average size and arrangement of the grains, or the strength of the electric field? Does one have to examine the switching domain before modeling any given HZO film?
(4) It is mentioned near the end of the manuscript that the authors examined the electric field near the domain wall of the film via FEA. But no further information was given regarding the setting of the FE model? Is there any complementary material in this regard?
Author Response
The authors do appreciate the comprehensive review and valuable remarks from Reviewer. We have tried to address each of the critical points and our answers to each comment are summarized below.
- I understand that the highlight of this study is the improved Lou’s statistical model which now takes into account the field inhomogeneity, depolarization field and charge injection. These are the aspects that are already believed to result in time dependence of polarization switching kinetics. What the authors did is to collect these formulations and to make the statistical model work for the observed low-amplitude switching behaviors. Of course it is always nice to see a systematic examination of the available options for modeling. It made me wonder what else novel contribution can be found through this investigation. Are there any other key mechanisms that can potentially be associated with such switching dynamics? Can this study lead to further insight into the polarization switching other than the aspects mentioned above? Is it really necessary to provide so many details associated with the models in 3.1 and 3.2? The authors have already published their studies regarding these two models, e.g., Ref. [4]. The contents in 3.1 and 3.2 are mostly identical to that published paper.
We are grateful for this comment. Let us split it in several parts.
I understand that the highlight of this study is the improved Lou’s statistical model which now takes into account the field inhomogeneity, depolarization field and charge injection. These are the aspects that are already believed to result in time dependence of polarization switching kinetics. What the authors did is to collect these formulations and to make the statistical model work for the observed low-amplitude switching behaviors. Of course it is always nice to see a systematic examination of the available options for modeling. It made me wonder what else novel contribution can be found through this investigation.
Indeed, this work is mainly aimed at developing a model that would take into account known physical phenomena that affect the process of polarization switching in HfO2. Refinement of the Lou’s model seemed to us the most suitable for this purpose. Unlike the most widely used NLS model, Lou’s equations allow to introduce electric field (total electric field of applied field, depolarization field and field of the injected charge) and thus they offer the opportunity to build a general model. We would like to note that the original Lou’s model does not work for HZO film, and consideration of specific properties of hafnia is required.
Are there any other key mechanisms that can potentially be associated with such switching dynamics?
We show that the electric field of the domain walls could act similarly to the depolarization field (Figure 6). For example, the electric field of the domain walls could accelerate polarization switching when the fraction of switched domains is less than 50%, and slow down switching, when the fraction exceeds 50%. Although this is quite different physical mechanism, its mathematical role in equations would be very similar to the depolarization field and thus we did not develop this idea.
Can this study lead to further insight into the polarization switching other than the aspects mentioned above?
We aimed to make our work useful for HfO2 applications and thus for FE-HfO2 memory developers. For example, the demonstrated way to introduce the total electric field to the polarization switching equations could be useful for developing an accurate way to calculate the retention loss in HfO2-based memory devices, which is now the main challenge for commercialization of HfO2-based memory.
From the fundamental point of view, we hope that our experimental verification of the predetermined character of polarization switching would be useful for the experts in the field.
Is it really necessary to provide so many details associated with the models in 3.1 and 3.2? The authors have already published their studies regarding these two models, e.g., Ref. [4]. The contents in 3.1 and 3.2 are mostly identical to that published paper.
Regarding chapter 3.1, we did use the NLS model with the Lorenz distribution in our previous work; however, we did not compare it with the classical NLS model. In addition, previously we did not take into account the contributions of the pulse trails to polarization switching and, accordingly, did not take into account the distortion they introduce into the experimental data. In the text of the new article, we directly point out the difference in the results associated with this factor: « In previous work [4], we indeed obtained a linear dependence of tLorentz on voltage in the range of small voltages, which is consistent with the results obtained in [22]. However, as can be seen from the plot in Figure 2d, the dependence obtained in this work for wider range of voltage is more similar to that obtained using the classical distribution function (2) for the NLS model [8]» (p. 6).
The content chapter 3.2 indeed echoes the content of the Ref [4], for example, we use the same equations to calculate the injected charge and imprint voltage. However, we believe that a brief description of the relevant theory is necessary to better understand the current paper for reader who may not be familiar with our previous work. Secondly, in a previous work, we focused on the impact of the information storage on the switching speed, and just mentioned that the read pulse could deviate the switching speed too. In current paper, we calculated the impact of the read pulse on the total field during the readout procedure and introduced it in model equations.
- The domain structures are shown in Fig. 4. However I did not see any grain morphology graphics of the HZO nanofilms. Apparently, the authors did not consider the polycrystalline microstructure that will affect the domain switching kinetics. It has been shown via phase-field modeling that the nature of the grain pattern in the nanofilms (e.g., grain size and orientation) and the existence of grain boundaries will affect the polarization switching or domain evolution dynamics in ferroelectric nanofilms, e.g., Su et al., Acta Materialia 87: 293-308, 2015; Zhang & Su, Int. J. Solids Struct. 254: 111939, 2022. The authors should discuss such findings in the context. In fact, as this study is regarding the modeling of polarization switching kinetics, the authors should widen the spectrum of their literature review by discussing microstructure-based approaches such as phase field modeling, which has been shown capable to describe time-dependence of polarization switching in ferroelectric nanofilms.
We are grateful for this comment. Unfortunately, PFM cannot provide unambiguous information about the effect of the grain pattern on polarization switching. However, it provides indirect information. We have added the relevant analysis to the text (p. 9-10, Figure 4). We are also grateful for the pointing out an alternative way of calculating the polarization kinetics, which takes into account the microstructure. We have added appropriate references to the overview of kinetics models in the introduction (p. 2).
- The extended statistical modeling of the polarization switching is based on a predetermined switching domain as demonstrated in Fig 4(a). How are the authors so sure that this assumption is applicable to all other polarization switching kinetics of polycrystalline HZO nanofilms? How reliable is assumption? What happens if other conditions change, say the thickness, the average size and arrangement of the grains, or the strength of the electric field? Does one have to examine the switching domain before modeling any given HZO film?
We agree that predetermined character of polarization switching revealed in 10 nm thick HZO should be additionally examined for films with other parameters, for example, for films with other thickness, other average size of grains and other structural texture. We have added the comment on this issue (p. 10).
- It is mentioned near the end of the manuscript that the authors examined the electric field near the domain wall of the film via FEA. But no further information was given regarding the setting of the FE model? Is there any complementary material in this regard?
We have added the details of FEA and improved an appropriate Figure (p. 12, Figure 6).

Reviewer 2 Report
The paper is devoted for polarization kinetics in thin films. The topic is generally interesting, however the paper contain unexplained places (below) and need major revisions.
Introduction should be expanded. List of references should be also expanded with references to new publications (2019-2022 years).
Lines 19-20 why You separate the contributions of the depolarization field and in-plane inhomogeneity of the electric field?
Line 116 the symbol Vrms should be explained.
The graphical quality of Fig. 4 should be improved.
Please explain why for investigations were selected HZO films with 10 nm thickness?
Conclusions should be rewritten in more informative way.
English need minor revisions.
Author Response
Response to Reviewer comments to our manuscript, entitled
“Polarization switching kinetics in thin ferroelectrics HZO films”
submitted to Nanomaterials, authored by E. Kondratyuk & A. Chouprik
(Manuscript ID: nanomaterials-1993330).
The authors do appreciate the comprehensive review and valuable remarks from Reviewer. We have tried to address each of the critical points and our answers to each comment are summarized below.
The paper is devoted for polarization kinetics in thin films. The topic is generally interesting, however the paper contain unexplained places (below) and need major revisions.
Below follows some additional suggestions and comments:
- Introduction should be expanded. List of references should be also expanded with references to new publications (2019-2022 years).
We are grateful for this comment. We have extended the discussion of previous art (p. 2).
- Lines 19-20 why You separate the contributions of the depolarization field and in-plane inhomogeneity of the electric field?
We have corrected this sentence as well as other similar sentences in the text.
- Line 116 the symbol Vrms should be explained.
Thank you for your comment. We have added a definition of Vrms: root mean square voltage (p. 4).
- The graphical quality of Fig. 4 should be improved.
Thank you. We have improved the resolution of this Figure by dividing it into three parts (Figures 4,5,6).
- Please explain why for investigations were selected HZO films with 10 nm thickness?
The comment on this issue has been added (p. 3).
- Conclusions should be rewritten in more informative way.
We are grateful for this advice. Conclusions has been rewritten (p. 13-14).
- English need minor revisions.
Thank you for your comment. We have tried to do our best.
Reviewer 3 Report
This is an interesting study and worthwhile for publication in the present form.
Author Response
We are grateful for such a high evaluation of our work by the Reviewer.
Round 2
Reviewer 1 Report
All my concerns were properly addressed. It is now considered ready for publication.
Reviewer 2 Report
Authors make proper corrections according to reviewer remarks and I suggest
to publish the paper as it is.